# Transcriptome analyses of *Anguillicola crassus* from native and novel hosts

Emanuel Heitlinger[1], Horst Taraschewski[2], Urszula Weclawski[2], Karim Gharbi[3] and Mark Blaxter[3,4]

[1] Department for Molecular Parasitology, Institute for Biology, Humboldt University Berlin, Berlin, Germany
[2] Department of Ecology and Parasitology, Zoological Institute, Karlsruhe Institute for Technology, Karlsruhe, Germany
[3] Edinburgh Genomics, The Ashworth Laboratories, The University of Edinburgh, Edinburgh, UK
[4] Institute of Evolutionary Biology, The Ashworth Laboratories, The University of Edinburgh, Edinburgh, UK

Corresponding author
Emanuel Heitlinger,
emanuelheitlinger@gmail.com

## ABSTRACT

*Anguillicola crassus* is a swim bladder nematode of eels. The parasite is native to the Asian eel *Anguilla japonica*, but was introduced to Europe and the European eel *Anguilla anguilla* in the early 1980s. A Taiwanese source has been proposed for this introduction. In the new host in the recipient area, the parasite appears to be more pathogenic. As a reason for these differences, genetically fixed differences in infectivity and development between Taiwanese and European *A.crassus* have been described and disentangled from plasticity induced by different host environments. To explore whether transcriptional regulation is involved in these lifecycle differences, we have analysed a "common garden", cross infection experiment, using deep-sequencing transcriptomics. Surprisingly, in the face of clear phenotypic differences in life history traits, we identified no significant differences in gene expression between parasite populations or between experimental host species. From 120,000 SNPs identified in the transcriptome data we found that European *A. crassus* were not a genetic subset of the Taiwanese nematodes sampled. The loci that have the major contribution to the European-Taiwanese population differentiation show an enrichment of synonymous and non-coding polymorphism. This argues against positive selection in population differentiation. However, genes involved in protein processing in the endoplasmatic reticulum membrane and genes bearing secretion signal sequences were enriched in the set of genes most differentiated between European and Taiwanese *A. crassus*. These genes could be a source for the phenotypically visible genetically fixed differences between European and Taiwanese *A. crassus*.

## INTRODUCTION

The precipitous decline of stocks of the European eel, *Anguilla anguilla*, over the last decades has spurred new research on these important fish (*Dekker, 2003a*; *Dekker, 2003b*).

While direct human influence such as overfishing and the destruction and damming of coastal habitats are undoubtedly the main reasons for the collapse of the eel population, the introduction of non-native pathogens may have contributed (*Sures & Knopf, 2004*).

The swim bladder nematode *Anguillicola crassus* was introduced from Asia to Europe early in the 1980s (*Kirk, 2003*; *Neumann, 1985*; *Taraschewski et al., 1987*). *A. crassus* is native to the Japanese eel *Anguilla japonica*, and has made a host jump to the European eel *An. anguilla*. A microsatellite study (*Wielgoss et al., 2008*), analysis of mitochondrial markers (*Wielgoss et al., 2008*; *Laetsch et al., 2012*) and historical reports (*Koops & Hartmann, 1989*) suggest that Taiwan was the most likely source of the founding population of the parasite, likely introduced by an import of live *An. japonica* eels to Northern Germany.

Adult *A. crassus* live inside the swim bladder of eels of the genus *Anguilla*. Female parasites shed eggs containing the L2 larval stage, which are released via the faeces into the water column. After hatching and ingestion by an intermediate host (copepods or ostracods; (*Moravec, Nagasawa & Miyakawa, 2005*)), L3 larval stages are infectious to the eel. When the eel host takes up infective L3, these migrate through the intestinal wall and the body cavity to the wall of the swim bladder, where they feed on tissue. After two additional moults (L3 to L4, and L4 to adult) sexually dimorphic adults enter the lumen of the swim bladder where they mate (*De Chaleroy et al., 1990*).

The parasite occurs at a higher prevalence in European eels than in Asian eels, and infects *An. anguilla* at higher infection intensities than *An. japonica*. Importantly, the parasite is more pathogenic to the European than to its native Asian host. While *An. japonica* mounts an immune response that eliminates many larvae, *An. anguilla* fails to mount such a response. The antibody response is delayed and insufficient (*Knopf, 2006*; *Knopf & Lucius, 2008*) and parasite larvae are not encapsulated in *An. anguilla* exposed to *A. crassus* (*Heitlinger et al., 2009*). *A. crassus* grows larger and produces more embryonated eggs in *An. anguilla* hosts compared to *An. japonica,* both in the wild (*Münderle et al., 2006*) and in laboratory experiments (*Knopf & Mahnke, 2004*). The inadequate immune response of *An. anguilla*, creating a more benign environment for the parasite, has been proposed to be the main reason for the altered dynamics of the host-parasite system (*Knopf, 2006*).

We are also interested in possible parasite contributions to these changed dynamics, and in particular in the possibility that the European eel-parasitising *A. crassus* have been selected in or have adapted to their new hosts. Under this model, nematode genetics would also contribute to the changed host-parasite dynamic. A genetic component of the differences between European and Asian *A. crassus* was identified in a cross-infection experiment under common garden conditions (*Weclawski et al., 2013*). European strains of *A. crassus* were found to differ in life history traits from ones sourced from Taiwan, independent of the experimental host species. In particular, European nematodes had an accelerated development compared to Asian nematodes (*Weclawski et al., 2013*). In the same experiment two European isolates from Poland were found to be diverged from those from Germany and Taiwan for morphological traits used to differenciate *Anguillicola* species (*Weclawski et al., 2014*).

We hypothesise that these genetic differences may result from allelic differences between the two nematode populations that either result in changed structures of key host-parasite interface effectors, or that these differences result in changes in expression of key genes involved in the interface. We assessed transcriptomic differences between age- and sex-matched European and Taiwanese *A. crassus* in a common garden, cross-infection design to disentangle differences in gene expression induced by intrinsic genetic differences of the nematodes from the influence of the host environment. The sequence data also allowed us to genotype the nematodes and test for non-neutral evolutionary processes influencing phenotypic and transcriptomic differences.

## METHODS

### Experimental infection of eels

L2 larvae used for the infection were collected from the swim bladders of wild yellow and silver eels from the River Rhine near Karlsruhe (49.0271N; 8.3119E) and from Lake Müggelsee near Berlin (52.4372N; 13.6467E) in Germany. Taiwanese larvae were obtained from eels from an aquaculture adjacent to Kao Ping River in south Taiwan (22.6418N; 120.4440E) and from a second aquaculture in Yunlin county (23.7677N, 120.2335E), approximately 150 km further north on the west coast of Taiwan.

*An. anguilla* were obtained from a farm in Northern Germany (Albe, Haren-Rütenbrock; 52.8383N; 7.1095E). *An. japonica* were caught at the glass-eel stage in the estuary of Kao-ping River (22.5074N; 120.4220E) and transferred to Germany. The absence of *A. crassus* before the experiment was confirmed in 8 *Anguilla japonica* and 4 *Anguilla anguilla.* After an acclimatisation period of 4 weeks (*An. anguilla*) or when they reached a size of >35 cm (*An. japonica*) eels were infected using a stomach tube. During the infection period water temperature was held constant at 20 °C. Eels were kept in 160 L tanks in groups of 5–10 individuals and provided with fresh, oxygenated water through continuous circulation. Eels were fed every two days with commercial fish pellets (Dan-Ex 2848; Dana Feed A/S Ltd, Horsens, Denmark) *ad libitum.*

At 55–56 dpi, eels were euthanized and dissected. The swim bladder was opened and after determination of their sex under a binocular microscope (Semi 2000; Zeiss, Oberkochen, Germany), adult *A. crassus* were immediately immersed in RNAlater (Quiagen, Hilden, North Rhine-Westphalia, Germany).

The experiment was approved by the Regierungspräsidium Karlsruhe approval no. 35-9185.81/G-120/06 and 35-9185.81/G-31/07.

### RNA extraction and preparation of sequencing libraries

RNA was extracted from 12 individual female nematodes and for 12 pools of 1 to 5 male nematodes using the RNeasy kit (Quiagen, Hilden, Germany) (see Table 1). The paired-end TruSeq RNA sample preparation kit (Illumina) was used to generate paired-end sequencing libraries with insert sizes of roughly 270 bp from polyA-selected RNA following the manufacturer's instructions. Multiple indexed paired-end adapters were used to enable multiplexing of the 24 different sequencing libraries in 3 pools of 8 samples each. These

**Table 1  Samples analysed for transcriptome response.**

| Sample name | Experimenal host species | Worm sex | Worm population | # worms prepared | Raw reads | Mapped reads | Analysed reads |
|---|---|---|---|---|---|---|---|
| AA_R11M | *An. anguilla* | male | Europe (Rhine) | 14 | 11,986,442 | 8,783,231 | 7,619,960 |
| AA_R16M | *An. anguilla* | male | Europe (Rhine) | 4 | 10,810,349 | 7,437,741 | 6,150,261 |
| AA_R18F | *An. anguilla* | female | Europe (Rhine) | 1 | 9,227,615 | 6,720,900 | 5,428,268 |
| AA_R28F | *An. anguilla* | female | Europe (Rhine) | 1 | 10,135,670 | 7,044,401 | 5,592,331 |
| AA_R2M | *An. anguilla* | male | Europe (Berlin) | 4 | 12,469,746 | 8,745,921 | 7,408,084 |
| AA_R8F | *An. anguilla* | female | Europe (Berlin) | 1 | 15,270,570 | 11,371,346 | 9,687,054 |
| AA_T12F | *An. anguilla* | female | Taiwan (KaoPing) | 1 | 11,299,438 | 8,196,168 | 6,727,218 |
| AA_T20F | *An. anguilla* | female | Taiwan (KaoPing) | 1 | 11,740,839 | 8,575,826 | 6,994,274 |
| AA_T24M | *An. anguilla* | male | Taiwan (KaoPing) | 3 | 8,552,723 | 6,023,322 | 5,053,565 |
| AA_T3M | *An. anguilla* | male | Taiwan (Yulin) | 4 | 11,031,751 | 7,783,403 | 6,730,362 |
| AA_T42M | *An. anguilla* | male | Taiwan (Yulin) | 1 | 11,573,501 | 8,013,752 | 6,829,319 |
| AA_T45F | *An. anguilla* | female | Taiwan (Yulin) | 1 | 10,646,847 | 7,554,730 | 6,314,234 |
| AJ_R1F | *An. japonica* | female | Europe (Rhine) | 1 | 9,855,005 | 6,983,544 | 5,814,315 |
| AJ_R1M | *An. japonica* | male | Europe (Rhine) | 1 | 10,211,903 | 6,951,868 | 5,828,185 |
| AJ_R3F | *An. japonica* | female | Europe (Rhine) | 1 | 9,897,937 | 7,100,162 | 5,618,547 |
| AJ_R3M | *An. japonica* | male | Europe (Rhine) | 2 | 8,775,211 | 5,981,163 | 5,006,069 |
| AJ_R5F | *An. japonica* | female | Europe (Berlin) | 1 | 11,949,105 | 8,814,614 | 7,562,071 |
| AJ_R5M | *An. japonica* | male | Europe (Berlin) | 1 | 11,231,532 | 7,859,814 | 6,651,999 |
| AJ_T19M | *An. japonica* | male | Taiwan (Yulin) | 7 | 9,195,576 | 6,605,467 | 5,733,247 |
| AJ_T20M | *An. japonica* | male | Taiwan (Yulin) | 8 | 10,862,591 | 7,715,619 | 6,437,571 |
| AJ_T25M | *An. japonica* | male | Taiwan (Yulin) | 5 | 11,195,315 | 7,565,845 | 6,416,480 |
| AJ_T26F | *An. japonica* | female | Taiwan (Yulin) | 1 | 11,195,335 | 8,051,694 | 6,833,011 |
| AJ_T5F | *An. japonica* | female | Taiwan (KaoPing) | 1 | 10,357,569 | 7,415,162 | 6,152,064 |
| AJ_T8F | *An. japonica* | female | Taiwan (Yulin) | 1 | 14,196,382 | 10,547,153 | 8,667,849 |

three pools all contained one random replicate each for each treatment combination ensuring complete statistical independence of replicates. The pools were sequenced on an Illumina Genome Analyzer IIX following the manufacturer's instructions. Raw data have been deposited in ENA under the study accession number SRP010338.

## De novo assembly, protein prediction and annotation

Trinity (version r2013-02-16) (*Grabherr et al., 2011*) was used to assemble raw sequencing reads into contigs representing transcripts and genes. Transdecoder (as supplied with Trinity) was used to predict protein coding genes. Based on these predicted proteins we obtained domain annotations using InterproScan (RC4) (*Zdobnov & Apweiler, 2001*) and sequence similarity using BLAST (*Altschul et al., 1997*) against SwissProt. Gene ontology (GO) terms were obtained either through association with domains in InterproScan (considered higher quality) or through assignment according to similarity (BLAST with a bitsore cutoff of 50, to increase annotation coverage).

We used the R-package topGO to traverse the annotation-graph and analyse each node in the annotation for over-representation of the associated term in focal gene-sets

compared to an appropriate universal gene-set with the "classic" method and Fisher's exact test (F-test). To test over-representation of Interpro domains we similarly used F-tests. The assembly contigs, read coverages and assignment to host, xenobiont and nematode groups, as well as contig sets identified through their differential expression are available for browsing in the online afterParty resource established for *A. crassus* at http://anguillicola.nematod.es.

## Mapping, abundance estimation and normalisation

All sequencing reads were mapped to the full Trinity assembly (including host and other contaminant contigs) using Bowtie version 2.1.0 (*Langmead & Salzberg, 2012*) and processed using RSEM (*Li & Dewey, 2011*) as indicated in the downstream analysis instructions of Trinity.

Briefly, ambiguously-mapping reads were assigned to the most appropriate transcript with RSEMs maximum likelihood method and a rounded counts (summed over technical replicates) for both the transcript and gene level were obtained. The final data used for all expression estimates was then calculated as fragments per kilobase of feature (transcript) per million fragments mapped (FPKM) based on trimmed mean of M values (TMM) normalisation (*Robinson & Oshlack, 2010*).

Genes and transcripts with less than 100 FPKM added over all samples were disregarded in further analyses. At this point we also excluded genes and transcripts of likely xenobiont (eukaryotic co-bionts of the nematode and fish, and laboratory contaminants) or host (eel, by comparison to a previous eel transcriptome (*Coppe et al., 2010*), and fish, from a taxonomic subset of the NCBI nr protein database) origin. We removed transcripts if BLAST hits (e-value cutoff 1e-5) against any of the fish or prokaryote databases were better than those against a nematode subset of nr.

## Analysis of expression data

We used multi-dimensional scaling (MDS), hierarchical clustering and $k$-means clustering to analyse the structure in complete expression data set as well as in male and female subsets. Based on these data we excluded two outlier samples ("AJ_T26F" and "AA_T42M"; The label is comprised of a two letter code for the host species [AJ|AA], a one letter code for the population [R|T], an arbitrary number and one letter for worm sex[F|M]).

The R-package edgeR (version 2.4.1) (*Robinson, McCarthy & Smyth, 2010*) was used to build negative binomial generalised linear models of expression. Models were based on a negative binomial distribution and the dispersion parameter for each transcript was approximated with a trend depending on the overall level of expression. In the maximal fitted model expression was regressed on nematode sex, host-species and parasite population, including all their interactions. The full model thus contained terms $Si + Hj + Pk + (SH)ij + (SP)ik + (HP)jk + (SHP)ijk + \epsilon$, where $\epsilon$ is the residual variance, $Si$ is the effect of the $i$th sex (male or female), $Hj$ is the effect of the $j$th host species (*An. anguilla* or *An. japonica*), $Pk$ is the effect of the $k$th population (European or Asian), $(SH)ij$ is the sex-by-species interaction and similarly for the other interactions.

The hierarchical nature of generalised linear models was respected considering (removing) all interaction effects of a main-term (e.g., $(SP)ik$, $(SH)ij$ and $(SHP)ijk$) when analysing models for the significance of that term (e.g., $Si$). Resulting $p$-values were corrected for multiple testing using the method of Benjamini and Hochberg and differential expression was inferred at a false discovery rate (FDR) of 5% (adjusted $p$-value of 0.05).

Alternatively we built the corresponding partial models with only the male and female subsets of the samples and estimated significance of host species and nematode population factors as before.

Random forests as implemented in the R-package RandomForest were used to additionally test for the ability to obtain a robust classifier separating host-species or nematode populations (and the combination of these factors) in decision trees on the gene expression data. We performed these tests on the full dataset and on subsets containing significant genes for focal contrasts in the GLMs.

## Identification of SNPs and genotype analysis

Samtools (version 0.1.18; mpileup) (*Li et al., 2009*) was used to call genotypes for individual nematodes and multi-nematode samples based on the Bowtie mapping used before for gene expression analysis. SNPs were filtered to have at least a phred-scaled quality of 30.

A matrix of genotypes was extracted for in which "0" coded homozygous reference, "1" heterozygous and "2" homozygous for the alternate allele. This matrix was read using the R-package adegenet (*Jombart, 2008*) and transformed to the other R-object types as needed for different packages.

Heterozygosity was calculated for individual nematodes using the R-package Rhh (*Alho, Välimäki & Merilä, 2010*). In addition to the relative heterozygosity we estimated internal relatedness (*Amos et al., 2001*), homozygosity by locus (*Aparicio, Ortego & Cordero, 2006*) and standardised heterozygosity (*Coltman et al., 1999*). F-statistics were calculated using the R-package hierfstat (*Goudet, 2005*) implementing the method of Weir and Cockerham (*Weir & Cockerham, 1984*) and Hardy–Weinberg-Equilibrium (HWE) for individual loci (SNPs) within populations was tested using the permutation method of the genetics package, as recommended for low sample sizes.

For multivariate analyses the genotype matrix was transposed to a distance matrix and analysed using neighbour joining and maximum parsimony trees with the R-package phangorn (*Schliep, 2011*). We then used principal component analysis (PCA) from the R-package adegenet (*Jombart, 2008*) to visualize the overall structure of the genotype data. The appropriate number of population clusters was estimated using $k$-means clustering of the first five principal components and analysis of the bayesian information criterion (BIC; function find.clusters). Discriminant analysis of principal components was then used to rank loci according to their contribution to the single remaining discriminant function between the two resulting groups (European vs. Asian).

We used a dn/ds threshold of 0.5 to assume positive selection. When whole genes with stretches potentially under different selection regimes are considered this has been suggested and used before (*Swanson et al., 2004*).

Kendall rank correlation tau tests were used to investigate correlations between different SNP, genotyping and expression statistics.

# RESULTS

## A common garden experiment

Populations of European *A. crassus* (sourced from the Rhine and Lake Müggelsee, Berlin, Germany) were compared to Taiwanese nematodes (sourced from two distinct aquaculture operations). They were used to infect both European *An. Anguilla* and Taiwanese *An. japonica* in a shared facility in a cross-infection experiment (similar to that of *Weclawski et al., 2013*). Adult nematodes were recovered, sexed and subjected to deep RNA-Seq analyses in a carefully randomized design. Adult female nematodes were large enough to be sampled individually, but RNA recovery from the smaller male nematodes meant that some male samples were pools of a small number (up to 5) of individuals taken from the same host eel. The RNA-Seq data were mapped to a transcriptome assembly, and after elimination of host transcript contamination, expression levels of nematode genes were compared between host species, sexes and treatments. The RNA-Seq data were also used to define and score single nucleotide polymorphisms (SNP) between the nematodes, and these genotyping data were used to explore the population genetics of the nematodes and their gene expression responses to infecting different hosts. Details of our methods and analyses are given in the Methods.

## More nematodes are recovered from matching host-parasite combinations

At the early time point of development (55–56 days post-infection (dpi)) chosen in our experiment we recovered more nematodes from the European population of *A. crassus* in *An. anguilla* and more of the Taiwanese population in *An. japonica*. This was true for both adult sexes of the nematode, as well as for L3 and L4 larval stages (Fig. 1; $p < 0.05$ for the interaction effects of host species and parasite population in generalized linear models). In geographically-matched host-parasite combinations, a mean of 7.8 (Taiwan) and 9.5 (Europe) of the 50 nematodes experimentally administrated were recovered as adults. For the cross-matched combinations of host species and parasite population, recovery of adult stage was on average only 1.4 (European nematode/*An. japonica*) and 4.4 (Taiwanese nematode/*An. anguilla*).

## Transcriptome assembly and annotation

We processed 12 individual female nematodes and 12 male nematode samples (batches of 1–5 individuals from one host) for Illumina RNA sequencing and obtained datasets of between $8.7 \times 10^6$ and $15.2 \times 10^6$ read pairs from each of the samples (Table 1). These reads were assembled into initial contigs representing 49,816 putative transcripts deriving from 33,173 transcript groups (or putative genes). These data have been made available for analysis in an afterParty resource (*Jones & Blaxter, 2013*) at http://anguillicola.nematod.es. These transcripts contain 60% (6788 of 11,372) of the previously deposited transcript reconstructions from a Roche 454 RNA-Seq experiment but were on average longer

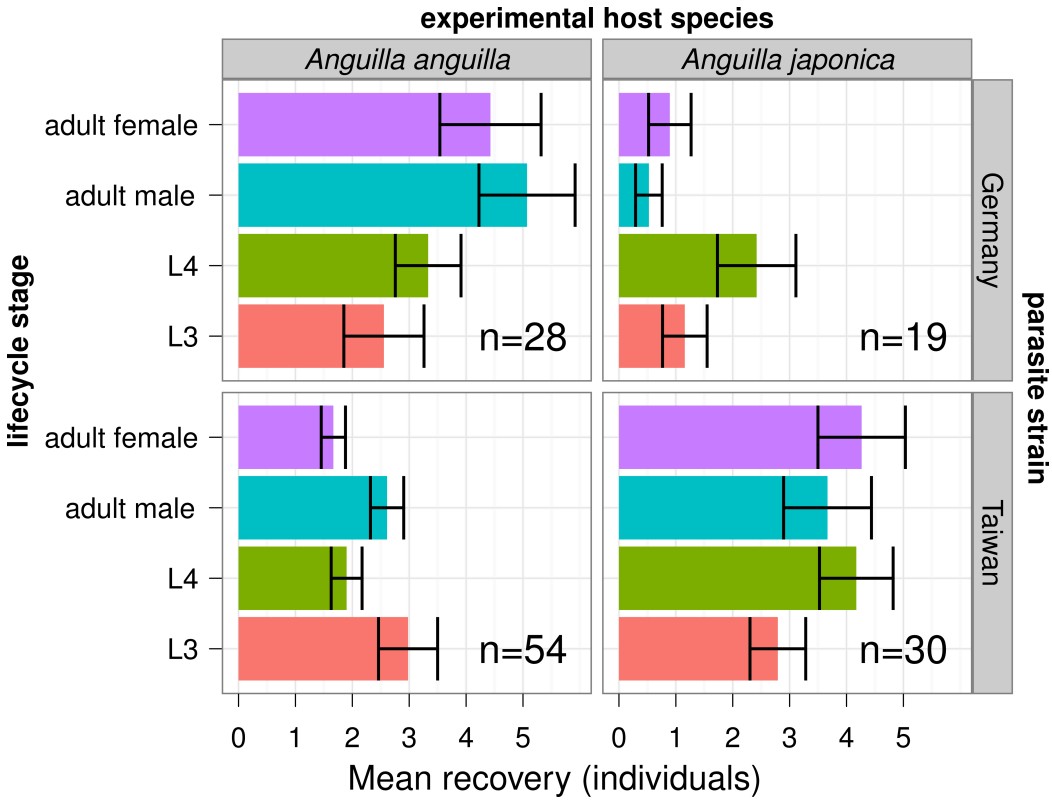

**Figure 1 Recovery of nematode populations in experimental host species.** *An. anguilla* and *An. japonica* were infected with a dose of 50 L3 stage larvae of *Anguillcola crassus* from European and Taiwanese populations. After 55–56 days different lifecycle stages of the nematodes were recovered and counted. Shown are the mean numbers of nematodes recovered from different host-parasite combinations, with errors bars indicating the standard error of these means. In host-parasite combinations occurring in the wild (European/European and Taiwan/Taiwan) more nematodes are recovered.

(median length 608 nt vs. 466 nt), more complete (68% vs. 12% including translation start codons) and covered a higher percentage of the proteome of the related nematode *Brugia malayi* (64% vs. 37%).

By applying stringent quality filtering for coverage (eliminating sequences reflecting only errors in the deep sequencing data) and taxonomic origin we selected a high-confidence *A. crassus* transcriptome that included 6,047 genes with 8,106 transcripts. This subset of transcripts was longer (median length 1794 nt) than the whole dataset, equivalently complete with respect to translation start codons (68%, as for the full set) and but had reduced coverage of the *B. malayi* proteome (51%). The majority of the raw sequence reads mapped to this high quality subset of the transcriptome assembly. The number of sequence reads analysed for expression and sequence polymorphism ranged from $5.0 \times 10^6$ to $9.7 \times 10^6$ per sample (Table 1).

After conceptual translation 6,633 (81.8%) of the transcripts were decorated with annotations based on protein similarity to SwissProt and 6542 (80.1%) with annotations based on InterPro domain signatures. For 5284 (65.2%) of the transcripts, Gene Ontology (GO) annotations were obtained through these domain signatures (Data S1).

## Gene expression differentiates sexes but not experimental hosts and parasite population

Multi dimensional scaling (MDS) of the overall expression data robustly grouped male and female nematode samples but failed to separate samples from different experimental hosts or by nematode geographical origin (Fig. S1A). Similarly female and male nematode samples clustered distinctly in hierarchical clustering of the overall expression data. The same clustering failed to differentiate samples from European and Asian experimental host species or nematode geographical origin (Fig. 2A). It was not possible to build a classifier grouping samples according to experimental host or parasite geographical population using random forests. The analysis prompted us, however, to exclude two samples ("AJ_T26F" and "AA_T42M") from further expression analysis based on their overall outlier expression profiles.

We identified 2154 (26.6%) of the transcripts as being significantly (FDR < 0.05; log fold-change > 1.5) differentially expressed between female and male nematodes (Data S1) based on generalised linear models taking into account all analysed factors (nematode sex, experimental host species and parasite population). The same models and thresholds recovered only very small sets of genes differentially expressed between the host species (27 transcripts; Data S2) and the parasite populations (30 transcripts, Data S3). These small sets of genes did not distinguish experimental host species or nematode populations in hierarchical clustering (Figs. 2B and 2C) or MDS analysis (Figs. S1B and S1C). Random forests also failed to find robust classifiers based on only these subsets of putatively differentially expressed genes.

## Coding sequence polymorphism and positive selection

We identified a panel of 128,707, bi-allelic, SNPs in 5008 genes over all nematode samples. The overall ratio of transitions to transversions rates (Ts:Tv) was 3.2. This can be expected in a transcriptome dataset due to a higher ratio of transitions in coding regions. We determined the effect of individual SNPs on coding sequence based on the conceptual translation and found 46,815 synonymous and 27,326 non-synonymous substitutions. The remaining 56,758 SNPs were in presumed untranslated regions (UTR), outside of open reading frames. The 13.28 SNPs per 1,000 sites comprised 27.36 synonymous SNPs per 1,000 synonymous sites and 5.49 non-synonymous SNPs per 1,000 non-synonymous sites. This resulted in an overall ratio of nonsynonymous substitutions per non-synonymous site over the synonymous substitutions per synonymous site (dn/ds) of 0.20.

Per-gene dn/ds was positively correlated with the total number of SNPs detected in (Kendall rank correlation tau test $p < 0.001$), so contigs with fewer SNPs also had a lower dn/ds on average. On the other hand the number of SNPs per gene was found to be positively correlated with the overall strength of gene expression (Kendall rank correlation tau tests $p < 0.001$). In contrast dn/ds was negatively correlated (Kendall rank correlation tau tests $p < 0.001$) with overall expression strength. Thus genes with higher overall expression had more SNPs but lower dn/ds, even while in general genes with more SNPs usually had a higher dn/ds.

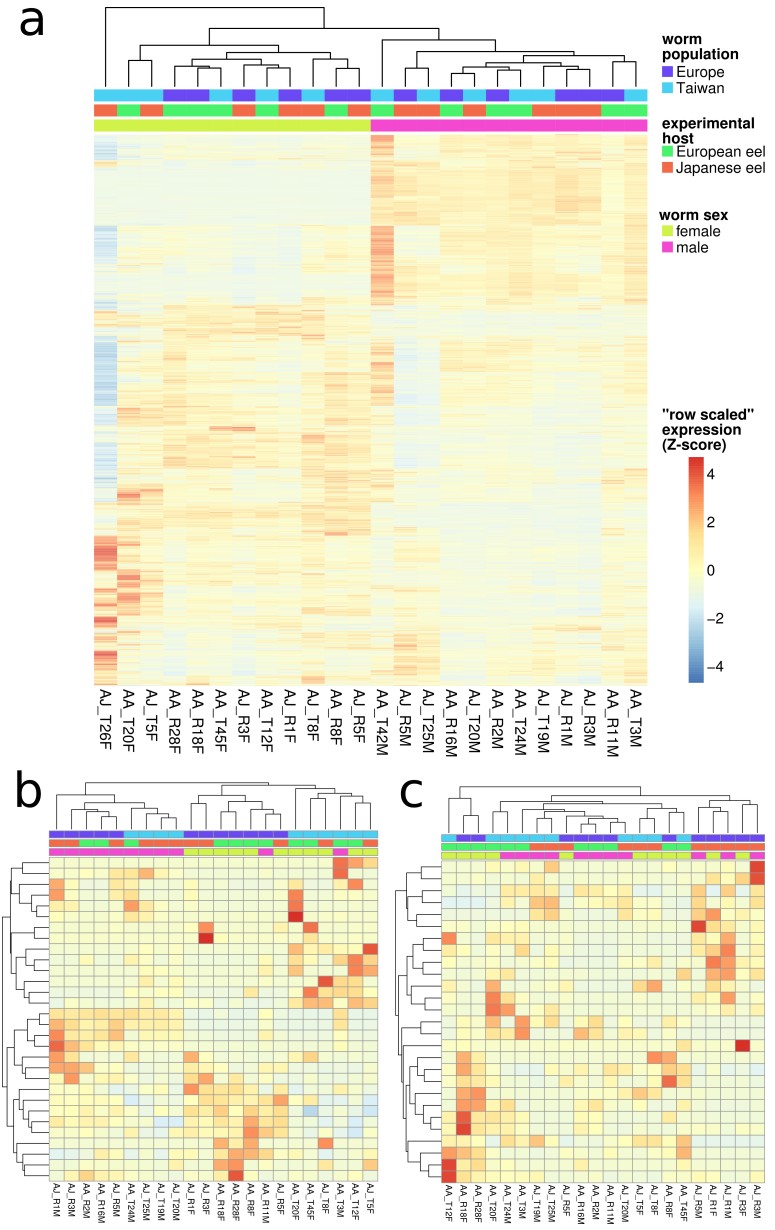

**Figure 2  Overall gene expression differences according to nematode population and experimental host species.** Gene expression changes in reconstructed transcripts were measured by mapping of sequencing reads. Normalized values were expressed in kilobase of feature (transcript) per million fragments mapped (FPKM) and plotted using the R-package pheatmap. Both rows (transcripts) and columns (nematode samples) were hierarchically clustered based on complete clustering of Euclidean distances. (A) depicts all 8,106 transcripts. In this analysis, the nematode samples are grouped correctly according to their sex. (B) shows the subset of 30 genes significantly differently expressed (FDR < 0.05; logFC > 1.5) between European and Asian nematode populations, and (C) the subset of 27 genes significantly differently expressed in the different experimental host species. Note that samples AJ_T26F and AA_T42M were removed because of their outlier status in the generalized linear models constructed for significance testing (with edgeR). They are thus also missing in (B) and (C) and all other gene expression statistics. While overall gene expression clearly distinguishes male and female nematodes, even the putatively differentially expressed gene-sets for nematode population or experimental host species cannot distinguish their respective contrasts.

Testing for GO term enrichment in the set of genes with high dn/ds ratios (dn/ds > 0.5) highlighted "zinc ion binding" and "protein binding" as over-represented molecular functions, "regulation of apoptotic process" and "cellular component biogenesis" as over-represented biological processes and "intrinsic to endoplasmic reticulum membrane" and "intracellular membrane-bounded organelle" as over represented cellular compartments among genes under positive selection. We note that these terms are all high-level GO terms and are thus relatively uninformative as to the shared functions of the selected gene set. This gene-set putatively under positive selection was not significantly enriched for signal sequences potentially leading to secretion (Fisher's exact test $p = 0.24$).

## European and Asian nematode populations are genetically differentiated

The European nematodes derived from an introduction, possibly of a small population, from a source in Asia, and would be expected to be genetically less diverse than the Asian nematodes, and nested within the diversity of the Asian nematodes. Inbreeding depression in a population with restricted diversity can result in an overabundance of homozygous genotypes. We inferred genotype statistics for the individual nematodes assayed in our experiment. We limited these heterozygosity based statistics to samples for which only one individual nematode was sequenced, as for pooled nematodes heterozygosity would obviously have been overestimated.

We detected no reduction of heterozygosity in European nematodes: neither relative heterozygosity (the ratio of heterozygous over homozygous genotypes), internal relatedness (*Amos et al., 2001*), homozygosity by locus (*Aparicio, Ortego & Cordero, 2006*) or standardised heterozygosity (*Coltman et al., 1999*) from these data indicated significantly higher heterozygosity in the Asian compared to the European nematodes (Table 2; one sided Mann–Whitney-Wilcoxon tests, $p > 0.05$). The observed relative heterozygosity was for all individuals higher than expected heterozygosity (0.173 in European, 0.175 in Taiwanese samples). The overall inbreeding coefficient $F_{IS}$ (the correlation of an individual's genotypes with genotypes found in European and Asian subpopulations) was negative ($-0.0544$), indicating that individual nematodes are less related than expected from a model of random mating within their population.

The overall fixation index ($F_{st}$) between European and Asian samples was 0.045, suggesting a rather low population structure. Nevertheless, a test for differentiation using G-statistics (*Goudet, 2005*; *Goudet et al., 1996*), indicated that it corresponds to highly significant genetic differentiation between populations. No significant differentiation was found within the European nematodes (between those sampled in the River Rhine and Berlin) or within Taiwanese nematodes (between the different sampling sites) as far as this could be analysed based on the low sample sizes for these subpopulations.

Tests for Hardy–Weinberg-Equilibrium (HWE) within populations were only possible for a subset (59%) of SNP markers and HWE could only be rejected ($p < 0.05$; in a permutation test) for 293 SNPs in the European population and 5407 SNPs in the Taiwanese population.

**Table 2** Heterozygosity measures for individual worms.

| Sample | Homozygous reference | Heterozygous | Homozygous alternate allele | Relative heterozygosity | Internal relatedness | Homozygosity by locus | Standardized heterozygosity |
|---|---|---|---|---|---|---|---|
| AA_R18F | 99,890 | 25,148 | 3,669 | 0.24 | −0.45 | 0.15 | 1.01 |
| AA_R28F | 98,624 | 26,049 | 4,034 | 0.25 | −0.42 | 0.16 | 1.00 |
| AA_R8F | 97,935 | 26,898 | 3,874 | 0.26 | −0.40 | 0.16 | 0.99 |
| AJ_R1F | 99,075 | 25,247 | 4,385 | 0.24 | −0.43 | 0.16 | 1.00 |
| AJ_R1M | 99,646 | 24,433 | 4,628 | 0.23 | −0.43 | 0.15 | 1.01 |
| AJ_R3F | 96,540 | 28,493 | 3,674 | 0.28 | −0.40 | 0.17 | 0.98 |
| AJ_R5F | 99,080 | 25,312 | 4,315 | 0.24 | −0.44 | 0.15 | 1.01 |
| AJ_R5M | 97,330 | 27,798 | 3,579 | 0.28 | −0.40 | 0.17 | 0.99 |
| AA_T12F | 97,278 | 27,141 | 4,288 | 0.27 | −0.38 | 0.17 | 0.99 |
| AA_T20F | 98,479 | 27,379 | 2,849 | 0.27 | −0.41 | 0.16 | 1.00 |
| AA_T42M | 99,514 | 23,700 | 5,493 | 0.23 | −0.41 | 0.15 | 1.01 |
| AA_T45F | 96,282 | 28,686 | 3,739 | 0.29 | −0.38 | 0.17 | 0.98 |
| AJ_T26F | 102,425 | 22,937 | 3,345 | 0.22 | −0.46 | 0.14 | 1.03 |
| AJ_T5F | 99,387 | 24,810 | 4,510 | 0.24 | −0.42 | 0.15 | 1.01 |
| AJ_T8F | 97,539 | 26,640 | 4,528 | 0.26 | −0.39 | 0.16 | 0.99 |

Because allele frequency based calculations could only be performed for nematodes that were sampled individually, we also used multivariate statistical analysis, which does not strictly depend on inference of heterozygosity and can therefore be used to analyse non-individual genotyping data (those missing from Table 2).

Population differentiation between nematodes sampled in Europe and Taiwan was also pronounced in this multivariate analysis as the distances between genotype matrices revealed a separation of genotypes from European and the Taiwanese populations. This differentiation was visible in both neighbour-joining and maximum parsimony trees computed on the distance matrix (Figs. 3A and 3B).

Further validation was provided by principal component analysis, in which the first component (explaining 12% of the total variance) separated nematodes from Taiwan and Europe clearly (Fig. 3C). The second principal component (explaining 9% of total variance) differentiated some of the European nematodes but did not show a clear pattern regarding origin (within Europe) or any other characteristics of the sampled nematodes.

## More synonymous polymorphism are found in genotypes distinguishing between populations

Clustering analysis (*k*-means) of principal components revealed a structure of only two clearly distinguishable groups in the data, identical to the European and Taiwanese samples. This was further validated by discriminant analysis of principal components (DAPC). The discriminant function (largely similar to principal component 1) permitted 100% accurate assignment of individual nematodes in bootstrapping tests to the correct source population, again demonstrating a clear differentiation between the European and Taiwanese samples (Figs. 4A and 4B).

**Peer**J

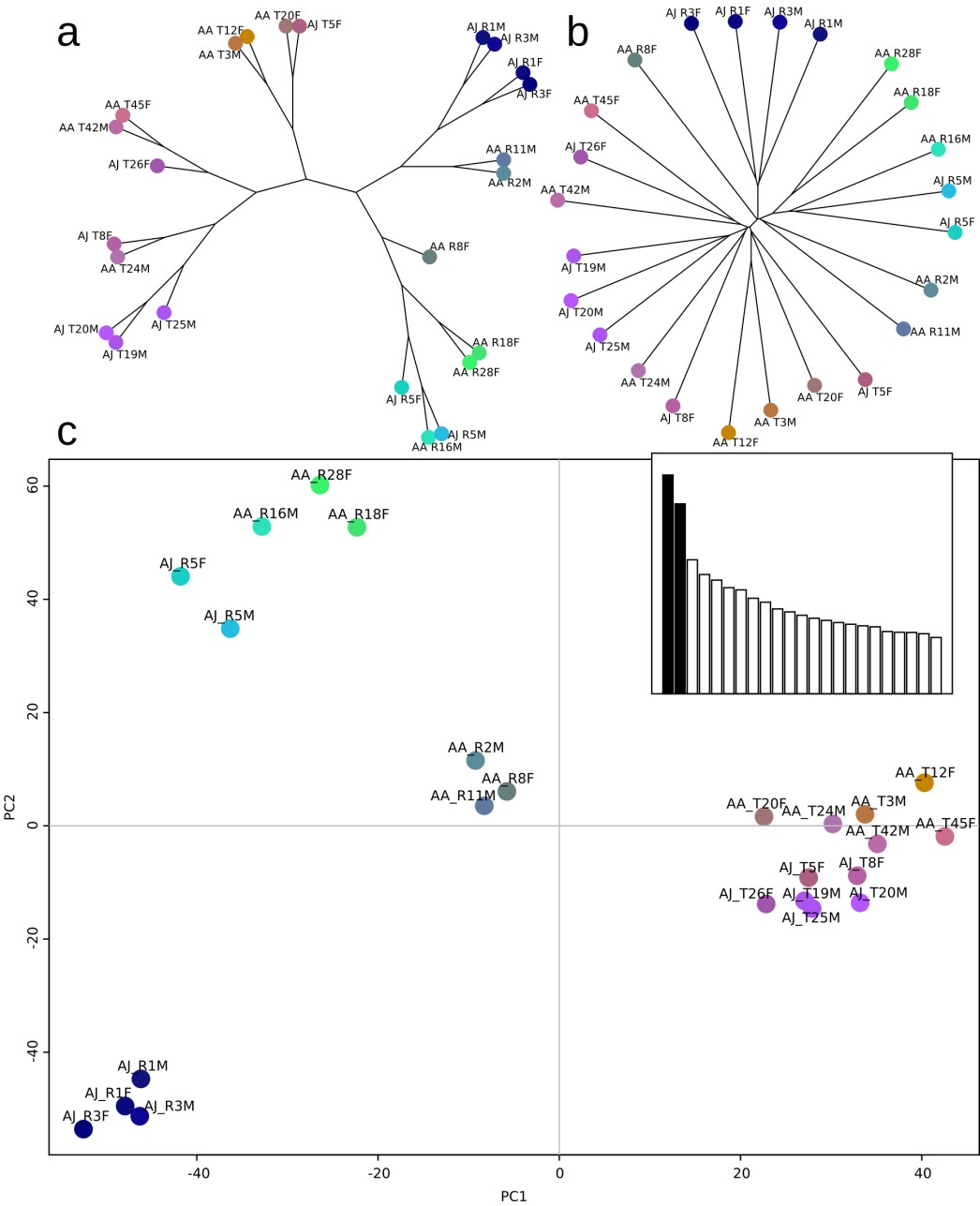

**Figure 3  European and Asian samples are differentiated by their genotypic profiles.** Based on mapping of sequencing reads to transcripts, 128,707 bi-allelic SNPs were called. Nematode samples were clustered into distinct clades transposing the genotype matrix to matrix of euclidean distances and analysing it with neighbour joining (A) and maximum parsimony (B) methods. Principal component analysis (PCA) was used to visualize the overall structure of the genotype data (C). The first principal component (PC1) explained 12% and the second component (PC2) 8% of the total variance in genotype differences according to the PCA eigenvalues (histogram inlay in C). PC1 clearly separates all Asian from all European nematode samples.

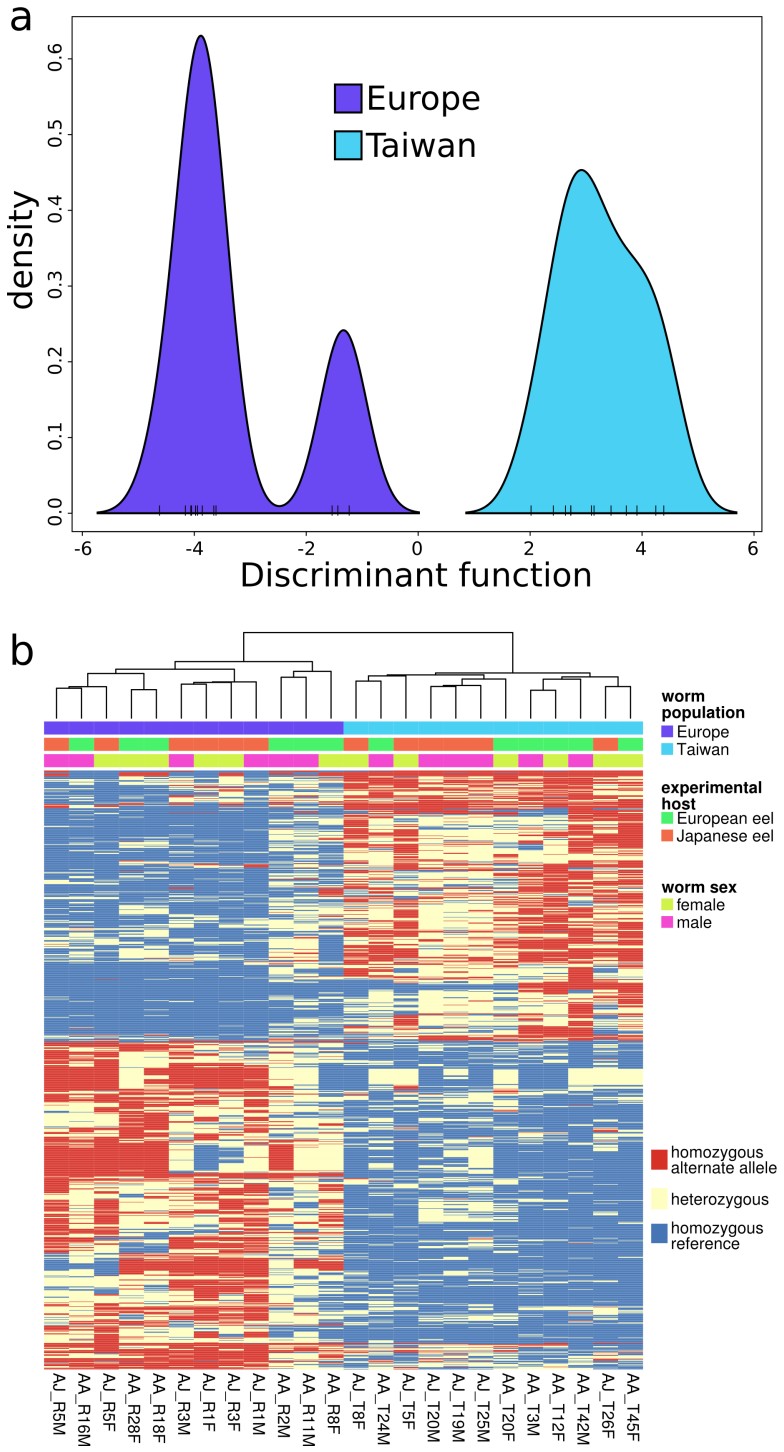

**Figure 4 Gene ranking for contributions to genotypic differentiation.** *K*-means clustering of the principal components identified only two clusters in the genotyping data corresponding to Asian and European nematode samples respectively. Discriminant analysis of principal components then estimated a discriminant function maximising the between-group variance. (A) displays the densities of individuals on the single retained discriminant function. This function was also used to rank loci according to their contribution to to genotypic discrimination. 589 loci with a contribution greater than $0.8 \times 10^{-5}$ are displayed in (B) and clustered based on Euclidean distances.

Genes associated with differentiation between nematode populations (maximal DAPC variable contribution $> 0.8 \times 10^5$ of a locus in the gene), were enriched for the GO terms "receptor signaling protein" and "exopeptidase" activity (molecular function), "endoplasmic reticulum membrane" and signal "peptidase complex" (cellular compartment). These genes were also significantly enriched for signal sequences leading potentially to secretion (Fisher's exact test $p < 0.001$).

The variable contribution of individual alleles to the discriminant function from DAPC was strongly correlated with $F_{st}$ (Fig. S2A), highlighting the agreement of both analyses in sorting loci for their contributing to the differentiation between the two nematode populations.

Loci with a stronger contribution to the differentiation between European and Taiwanese nematodes showed an overrepresentation of synonymous SNPs and SNPs in UTRs (Fig. S2B). All measures of differentiation used ($F_{st}$, $F_{is}$, contribution to the discriminant function and PC1 loading) were thus significantly higher for polymorphisms without an effect on the protein sequence (Mann–Whitney-Wilcoxon tests, $p < 0.001$).

## Differential gene expression between sexes is negatively correlated with genotypic differentiation between populations

Both the mean and maximal signals for genotypic separation in populations over loci per gene ($F_{st}$, contribution to discriminant function, loading of PC1) were negatively correlated with absolute values of log fold-change between male and females. Similarly the $p$-values for expression differences between sexes were positively correlated with all measurements of genotypic differentiation. Thus genes expressed more differentially between sexes were associated with lower genotypic differentiation between populations. This is especially striking as higher expression was not associated with genotypic differentiation overall.

## DISCUSSION

Common garden experiments are a classical method in evolutionary biology to disentangle genetic from environmentally induced effects. A cross infection experiment is the obvious extension of this approach to a host-parasite system.

We used a common garden experiment to demonstrate that *A. crassus* shows an "adapted" ability to successfully infect *An. anguilla* and *An. japonica*, as more parasites were recovered in matching infections (European parasites in *An. anguilla*; Taiwanese parasites in *An. japonica*). These differences are in good agreement with previous data at an earlier time of infection in a similar experiment (*Weclawski et al., 2013*). As noted before, this ("adapted") pattern does not necessarily reflect adaptation, as we cannot assume that earlier development leads to higher parasite fitness, which would better be measured by lifetime reproductive success.

We analysed parasite geographical source and host environment induced differences on gene-expression using transcriptomics. We did not detect differences in gene expression between the parasites infecting *An. japonica* and *An. anguilla*. The gene expression profiles of young adult stages of *A. crassus* seem inert to the environment imposed by the different

host species. This is unexpected as large morphological differences are observed between nematodes from different host species in the wild (*Münderle et al., 2006*) and in laboratory infections (*Knopf & Mahnke, 2004*; *Weclawski et al., 2013*; *Weclawski et al., 2014*). A possible reason might be that nematodes are influenced by and respond to the host immune system only during the larval stages migrating through tissues. The transcriptome of haematophagous adult stages living in the swim bladder may be rather unaffected by the host environment, and the phenotypic responses (parasite size) a result of improved larval health in the compromised host. It is possible that our experimental design might have selected nematodes with a "healthy" transcriptome, one that allows survival irrespective of the experimental host. This "healthy" transcriptome might then be largely the same in different host environments.

We identified no transcriptomic signature of the differences in life history traits between European and Taiwanese nematode populations that have been reported from cross-infection experiments irrespective of the host environment (*Weclawski et al., 2013*). The faster development of the European population of A. *crassus* thus has no obvious correlation with an early adult expression phenotype.

A previous report of tentative gene expression differences between single European and Taiwanese nematodes in their respective natural hosts (*Heitlinger et al., 2013*) may be explained as a product of stochastic noise in transcriptome sampling without repetition. The sampling size used for the present study (24 nematodes, six per treatment group) is relatively high for a transcriptomics study and, while it is never possible to prove a negative, we consider the present negative results to be true negatives.

As an argument supporting this notion, our analysis was able to find sex dimorphic expression differences in roughly one third of the analysed genes. This difference corresponds to a good resolution according to roughly one to two third of genes showing differences between sexes throughout many animal taxa (including studies with even higher repetition) (*Cutter & Ward, 2005*; *Jin et al., 2001*; *Yang et al., 2006*).

As high as it is for a transcriptomic experiment, the sample size of 24 nematodes from two populations is very small for a population study. Nevertheless, we demonstrate that a combination of classical and multivariate analysis can be useful for population genetic screening of such a dataset originally obtained for analysing gene expression. High-density genotyping of nematodes from our transcriptome sequencing demonstrated that the European parasite population is not a genetic sub-sample of the sampled Taiwanese population. The clear differentiation between nematodes from Europe and Taiwan, but not within sampling sites on the Taiwanese East cost and from Southern and North-Eastern Germany indicates that the European population of A. *crassus* might have a genetically distinct origin from our Taiwanese isolates.

The coefficient of population differentiation $F_{st}$ (*Wright, 1949*) between European and Taiwanese individuals had a value of 0.045 and could be interpreted as indicating negligible differentiation. Nevertheless, we show that, based on tests using g-statistics and multivariate analysis, the deep set of SNPs is fully sufficient to assign nematodes to European or Asian populations without any error. These pairwise $F_{st}$ values between

Taiwanese and European samples are in line with previous findings of *Wielgoss et al. (2008)*, who observed values between 0.02 and 0.056.

DAPC permitted the measurement of the degree of differentiation for both genes and samples. The contribution of genes to the discriminant function is highly correlated with $F_{st}$, but has advantages for low sampling sizes and ultra-deep sampling of markers throughout the genome (*Jombart, Devillard & Balloux, 2010*).

We can also conclude from low values of the overall inbreeding coefficient $F_{is}$ that the European populations of *A. crassus* show no heavy signature of inbreeding after a genetic bottleneck. Additional evidence for this is provided by high values of heterozygosity for individual European nematodes, which are not reduced compared to Taiwanese individuals. This result is largely in line with a previous study, which reported an universally high heterozygosity in Northern Europe, using larger sample sizes (*Wielgoss et al., 2008*). The same authors observed only marginally higher heterozygosity in isolates from Taiwan.

In the present study, however, we did not analyse Taiwanese nematodes from free-living eels. Individual *A. crassus* from wild caught eels displayed higher heterozygosity than those sampled from aquaculture operations in our own studies before (*Heitlinger et al., 2013*) and thus some additional genotypic diversity is likely to be found within Taiwan.

Populations from Taiwanese aquacultures might not be in Hardy–Weinberg Equillibrium, as this might only fail to be rejected in most markers due to the low sample sizes. It cannot be excluded that Taiwanese isolates are from populations experiencing a strong Wahlund effect, as observed in isolates from the River Rhine (*Wielgoss et al., 2010*). A continued mixing of population could have resulted in the affiliation of our two different Taiwanese isolates over the European isolates.

In the absence of gene flow within approximately 100 generations since its introduction, *A. crassus* populations could have undergone substantial genetic drift, explaining the clear distinction from the presumed Taiwanese source populations. Alternatively, the Taiwanese populations may not be so closely related to the actual source of European *A. crassus*, and Asian populations of *A. crassus* more closely related to the true source population of the European isolates may be identified in future studies involving larger sample sizes. In the meantime we encourage caution in discussion of genetic differences in European and Taiwanese isolates of *A. crassus* as having been induced by their translocation (i.e., selection by host or environment), as the same differences may already be present between unsampled Asian isolates.

For our present study the patterns of SNPs in protein-coding genes of *A. crassus* is informative. The estimated dn/ds of 0.2 is only slightly lower than the 0.244 previously obtained from 454 pyrosequencing of the transcriptome (*Heitlinger et al., 2013*). As expected, genes with overall higher gene expression had a lower dn/ds, probably because genes with higher expression are under stronger purifying selection (*Drummond et al., 2005*).

Our previous analyses identified proteinases as possibly being under positive selection (*Heitlinger et al., 2013*), but we did not observe enrichment of for proteinases in the set of putatively positively selected genes (using the same threshold of dn/ds > 0.5). The reason

for this is mainly additional non-coding SNPs observed in the respective proteinase genes (see below why some peptidases still show an interesting pattern of genotypes).

Merging population differentiation estimates with information on coding polymorphism we found that loci contributing to the differences between European and Taiwanese *A. crassus* were enriched for synonymous polymorphisms. These loci therefore do not show evidence of positive selection. Nevertheless genes associated with loci that discriminated between populations showed enrichment for functional categories that might be important for host-parasite interaction, and variation in these genes could explain differences between Taiwanese and European *A. crassus*.

Signal sequences that direct newly translated proteins to enter the secretory system of the endoplasmic reticulum were found more often than expected in genes that differentiated between populations. Additionally signal peptidase-associated processing at the endoplasmic reticulum membrane and general endopeptidases showed high differentiation between *A. crassus* from Taiwan and Europe. Interestingly three of these endopeptidases were among the twelve peptidases reported previously to possess a high dn/ds and to be potentially under positive selection (*Heitlinger et al., 2013*).

While the present study does not find an excessively high dn/ds in these peptidases, these data are suggestive of a key role of secreted peptides and their processing in the differences between *A. crassus* from Europe and Taiwan.

## ACKNOWLEDGEMENTS

We thank Yun-San Han and his group at National Taiwan University for support obtaining Japanese Eels and *A. crassus* larvae. We thank the staff of Edinburgh Genomics for processing samples. We thank Sèbastien Wielgoss for comments on the manuscript and analyses.

### Funding

This work was funded by Volkswagen Foundation in the "Förderinitiative Evolutionsbiologie" (grant to Emanuel Heitlinger). Edinburgh Genomics is partly supported through core grants from NERC (R8/H10/56), MRC (MR/K001744/1) and BBSRC (BB/J004243/1). The funders had no role in study design, data collection and analysis, decision to publish, or preparation of the manuscript.

### Grant Disclosures

The following grant information was disclosed by the authors:
Volkswagen Foundation.
NERC: R8/H10/56.
MRC: MR/K001744/1.
BBSRC: BB/J004243/1.

### Competing Interests

The authors declare there are no competing interests.

## Author Contributions

- Emanuel Heitlinger conceived and designed the experiments, performed the experiments, analyzed the data, contributed reagents/materials/analysis tools, wrote the paper, prepared figures and/or tables, reviewed drafts of the paper.
- Horst Taraschewski conceived and designed the experiments, contributed reagents/materials/analysis tools, reviewed drafts of the paper.
- Urszula Weclawski performed the experiments.
- Karim Gharbi conceived and designed the experiments, contributed reagents/materials/analysis tools.
- Mark Blaxter conceived and designed the experiments, contributed reagents/materials/analysis tools, wrote the paper, reviewed drafts of the paper.

## Animal Ethics

The following information was supplied relating to ethical approvals (i.e., approving body and any reference numbers):

The experiment was approved by the Regierungspräsidium Karlsruhe approval no. 35-9185.81/G-120/06 and 35-9185.81/G-31/07.

## DNA Deposition

The following information was supplied regarding the deposition of DNA sequences:

Raw reads are deposited in ENA under the study accession number SRP010338.

## Data Deposition

The following information was supplied regarding the deposition of related data:

The transcriptome assembly is deposited in the online afterParty resource established for *A. crassus* at http://anguillicola.nematod.es.

## Supplemental Information

Supplemental information for this article can be found online at http://dx.doi.org/10.7717/peerj.684#supplemental-information.

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
