# Peer review of "Transcriptome analyses of Anguillicola crassus from native and novel hosts"

_PeerJ, doi:10.7717/peerj.684_

## Round 0.1 · original submission · Minor Revisions

Dear Authors,

The reviewers have suggested some minor revisions and suggestions for your manuscript.

Reviewer 1 ·

Basic reporting

This manuscript reports new and interesting findings on genetic/transcriptomic differentiation of A. crassus in Taiwan and Europe in the host species A. anguillae and A. japonica.
The manuscript is well written and understandable. The introduction gives a good understanding of the background and idea of the study with sufficient literature cited.

Experimental design

No comments

Validity of the findings

The data is analyzed appropriately and presented transparently. Conclusions are drawn with care and possible problems related to study design, no. of replicates etc. are discussed.

Additional comments

Some minor comments:
The first three sentences in the Abstract are not very well connected. Maybe the authors can make this part a bit more "fluent" to read.

Reviewer 2 ·

Basic reporting

No comment

Experimental design

The methods should include more details of how GO terms were assigned e.g. software, version numbers, datasets, BLAST cut-offs. Also were there any non-default settings used for Trinity etc. There was no reference for TopGO or Interproscan.

How was the contaminating sequence removed? Using BLAST, Kraken etc? What were the parameters?

Validity of the findings

No comment

Additional comments

It would be useful to mention earlier in the abstract that the parasite is a nematode.

Non-native should be hyphenated (l18).

Moulds should be moults (l35)

The text states that “Genes and transcripts with less than 100 FPKM over all samples…”. Does this mean that the FPKM had to be >100 in every sample or when FPKMs were added up for every sample, this value had to be >100?

The outlier samples which were removed could do with more explanation, e.g. what conditions were these from?

The improved reliability of convergence of EdgeR over DESeq (DESeq2?) could do with more explanation or be left out. I do not find it a problem you using one rather than the other, but the partial explanation leaves me wondering what tests you did. I would be happy if you didn’t say why you picked EdgeR.

Was there no additional filtering of SNPs? Were there not problems with sequencing depth in some cases as RNA-seq coverage could be very high. I believe some SNPs callers will have problems when there is very high coverage.

Figure 1b should be Figure S1b (l283).

Again there is no explanation of what the sample ids mean in Figure S1.

L302 – what does reliable mean here?

When stating the ratio of transitions to transversions (l303) you should give the reader some idea of your interpretation of this result.

What is your interpretation of the results in lines 313-321? If there is no conclusion from this then it could be lost from the manuscript.

Is 0.5 a good dN/dS cut off for calling positive selection? I don’t think you can infer positive selection unless it is over 1.

When inferring GO terms largely from interpro you will always have problems with very general terms coming up. Were there any individual genes or large gene families (not annotated with particular GO terms) which had a high dN/dS?

L369 misspelling of Taiwanese

In their discussion the authors state that in their natural hosts the parasites produce more offspring and are therefore better adapted. I would argue that this is not necessarily true. If greater fitness is associated with greater pathology it may be that the parasites are less well adapted, as well adapted parasites usually cause little pathology. What is the pathology in each part of the common garden experiment?

There are some additional errors in language.

·

Basic reporting

The article is very well written. I only have a few minor points.

Line 35: I think ‘moulds’ should be moults.

Line 423: I think ‘as’ is missing from this sentence.

Figure 1 legend: the species name Anguillcola crassus is not italicised.

Experimental design

The experiments are well designed. Again, I only have a few minor points.

The eels are infected with Anguillicola crassus using a stomach tube. Presumably the authors are confident the recipient eels are uninfected? I noticed the Weclawski et al 2013 paper (that describes a similar ‘common garden’ experimental infection) states: ‘The absence of A. crassus was confirmed by dissection of 10 randomly chosen eels of each species.’ Did the authors perform the same test?

Validity of the findings

The authors perform a population study using a low sample size. Obviously, the data was obtained for analysing gene expression and the authors are maximising the information contained within the sequence data. I’m not in a strong position to comment on the extent to which the low sample sizes constrain any conclusions. However, as the authors clearly confront this issue in the article and are appropriately cautious when discussing the results, I believe PeerJ readers can come to their own conclusions.

Additional comments

Just a thought while reading the discussion.

The authors find no significant differences in gene expression between parasite populations or between experimental host species even in the face of clear phenotypic differences in life history traits. In the discussion, it is suggested that the

'nematodes are influenced by and respond to the host immune system only during the larval stages migrating through tissues. The transcriptome of haematophagous adult stages living in the swim bladder may be rather unaffected by the host environment, and the phenotypic responses (parasite size) a result of improved larval health in the compromised host.'

This is entirely plausible but could other transcriptomic differences (present in the adult but perhaps not detectable using the techniques applied) also be important? For example, could critical parasite genes display altered patterns of alternative splicing in different hosts, so that even though there is no significant difference in the overall expression of a gene, the ratios of functionally distinct isoforms are being altered. In such a scenario, subtle changes to the expression of a small number of splicing regulators could possibly account for clear phenotypic differences observed. I don't know if the data allows interrogation of different splice patterns but such transcriptomic changes may be worth considering.

---

## Round 0.2 · accepted · Accept

Dear Authors,

Thank you for making those minor corrections to the manuscript.